# Comparative genome analysis of test algal strain NIVA-CHL1 (*Raphidocelis subcapitata*) maintained in microalgal culture collections worldwide

**Takahiro Yamagishi**[1]*, **Haruyo Yamaguchi**[2◦], **Shigekatsu Suzuki**[2◦],
**Mayumi Yoshikawa**[3◦], **Ian Jameson**[4‡], **Maike Lorenz**[5‡], **David R. Nobles**[6‡],
**Christine Campbell**[7‡], **Masanori Seki**[3‡], **Masanobu Kawachi**[2‡], **Hiroshi Yamamoto**[1‡]

**1** Ecotoxicity Reference Laboratory, Risk Assessment Science Collaboration Office, Center for Health and Environmental Risk Research, National Institute for Environmental Studies (NIES), Tsukuba, Ibaraki, Japan, **2** Center for Environmental Biology and Ecosystem Studies, National Institute for Environmental Studies (NIES), Tsukuba, Ibaraki, Japan, **3** Chemicals Evaluation and Research Institute, Japan (CERI), Kurume, Fukuoka, Japan, **4** Australian National Algae Culture Collection (ANACC), Commonwealth Scientific and Industrial Research Organisation (CSIRO), Castray Esplanade, Hobart, Tasmania, Australia, **5** Culture Collection of Algae at Göttingen University (SAG), Georg-August-University Göttingen, Göttingen, Germany, **6** The University of Texas at Austin, Austin, Texas, United States of America, **7** Culture Collection of Algae and Protozoa (CCAP), Scottish Association for Marine Science (SAMS), Oban, Argyll, United Kingdom

◦ These authors contributed equally to this work.
‡ These authors also contributed equally to this work.
* yamagishi.takahiro@nies.go.jp

**Data Availability Statement:** The sequence data were deposited in DDBJ under accession number DRA010570 (DRR238908-DRR238916).

## Abstract

*Raphidocelis subcapitata* is one of the most frequently used species for algal growth inhibition tests. Accordingly, many microalgal culture collections worldwide maintain *R. subcapitata* for distribution to users. All *R. subcapitata* strains maintained in these collections are derived from the same cultured strain, NIVA-CHL1. However, considering that 61 years have passed since this strain was isolated, we suspected that NIVA-CHL1 in culture collections might have acquired various mutations. In this study, we compared the genome sequences among NIVA-CHL1 from 8 microalgal culture collections and one laboratory in Japan to evaluate the presence of mutations. We found single-nucleotide polymorphisms or indels at 19,576 to 28,212 sites per strain in comparison with the genome sequence of *R. subcapitata* NIES-35, maintained at the National Institute for Environmental Studies, Tsukuba, Japan. These mutations were detected not only in non-coding but also in coding regions; some of the latter mutations may affect protein function. In growth inhibition test with 3,5-dichlorophenol, EC50 values varied 2.6-fold among the 9 strains. In the ATCC 22662–2 and CCAP 278/4 strains, we also detected a mutation in the gene encoding small-conductance mechanosensitive ion channel, which may lead to protein truncation and loss of function. Growth inhibition test with sodium chloride suggested that osmotic regulation has changed in ATCC 22662–2 and CCAP 278/4 in comparison with NIES-35.

**Funding:** This study was supported by an internal funding from National Institute for Environmental Studies (NIES) to TY (There is no specific number for this grant.), and "partially" funded from the National BioResource Project (NBRP) from Japan Agency for Medical Research and Development (AMED): we declare there was no additional external funding received for this study. Also, we declare the funders had no role in study design, data collection and analysis, decision to publish, or preparation of the manuscript.

**Competing interests:** The authors have declared that no competing interests exist.

## Introduction

*Raphidocelis subcapitata* (= *Pseudokirchneriella subcapitata*, *Selenastrum capricornutum*) is a sickle-shaped, freshwater green microalga from the Selenastraceae family. It is one of the most frequently used species in algal growth inhibition tests; some public guidelines such as those of the Organisation for Economic Co-operation and Development (OECD) and United States Environmental Protection Agency (USEPA) recommend this species because of its high growth rate, sensitivity to toxicants, and good reproducibility in comparison with those of other algae [1–2]. Now, many microalgal culture collections worldwide maintain *R. subcapitata* for distribution to users.

All strains of *R. subcapitata* in these collections are derived from the same cultured strain, NIVA-CHL1. This strain was isolated from Nitelva river, Akershus, Norway by O.M. Skulberg in 1959 and deposited to microalgal culture collections worldwide to be used as a standard green algal strain for toxicity assays. Since the OECD eventually adopted the algal growth inhibition test using this strain in 1981, it has been used in some research fields such as ecotoxicology and evaluation of environmental risk of chemical substances. In each culture collection, it has a specific deposition number, e.g. ATCC 22662, CCAP 278/4, UTEX 1648, SAG 61.81 and NIES-35.

Suzuki et al. (2018) sequenced the nuclear, mitochondrial, and plastid genomes of *R. subcapitata* NIES-35 and compared them with those of other algal species to assess genome evolution and to understand the role of genetic background in environmental adaptation and the mode of action of toxicants. Phylogenetic analysis based on plastid genome sequences suggested that *R. subcapitata* is in the most basal lineage of the four studied species in Selenastraceae, indicating its early divergence from a common ancestor of the four species [3]. The mitochondrial genome shows dynamic evolution history, with intron expansion in the Selenastraceae. The nuclear genome encodes 13,383 proteins, and is the smallest (51 Mbp) among those of the closely related Sphaeropleales species (68 Mbp in *Monoraphidium neglectum*, 108 Mbp in *Tetradesmus obliquus*, 58 Mbp in *Chromochloris zofingiensis*, and 111 Mbp in *Chlamydomonas reinhardtii*) [3]. The genome of *R. subcapitata* encodes an $H^+$/hexose cotransporter that is functionally related to glucose intake, indicating that *R. subcapitata* might grow mixotrophically using exogenous glucose. The genome of *R. subcapitata* also encodes multiple nicotianamine transporters. Nicotianamine is an organic chelator of various metals that the higher plants also possess. The presence of these genes indicates that *R. subcapitata* might take up metals into the cell not only as free ions but also as complexes formed by nicotianamines it secretes into the environment. Thus, the genome sequence of *R. subcapitata* has revealed some important physiological characteristics of this species and its phylogenetic position.

No full genome sequences of strains other than NIES-35 have been obtained. To evaluate the phylogenetic position of NIVA-CHL1, the rubisco large subunit (*rbc*L) region of ATCC 22662 has been sequenced [4]. In our previous study, our alignment of the *rbc*L regions between NIES-35 and ATCC 22662 showed a mismatch of two base pairs in the *rbc*L gene [5]. Although sequencing errors might explain the mismatch, we suspected that NIVA-CHL1 strains maintained in different collections under different conditions have accumulated some mutations over the 61 years that have passed since NIVA-CHL1 was isolated from the natural environment. Genome mutations are well known to accumulate following long-term sub-culturing because artificial selection of variants occurs when part of the old culture is transferred to a new culture vessel.

In this study, we sequenced the whole nuclear genomes of NIVA-CHL1 strains from 8 microalgal culture collections and one laboratory in Japan to evaluate the mutations accumulated over 61 years. In addition, it is critical issue for the standard strain whether sensitivities

to toxicants vary among the strains maintained at culture collections worldwide. To assess the variation in sensitivity to toxicants, we compared the toxicity values (EC50, EC10, and no-observed-effect concentration [NOEC]) of the reference substance 3,5-dichlorophenol (3,5-DCP) among the 9 strains. Also, we attempted to identify the critical mutations to change physiological characteristics with physiological approaches. Finally, we discuss the appropriate maintenance method of the test algal strain in an algal culture collection or laboratory for preserving its genetic homogeneity and the reliability of test results.

## Materials and methods

### Cultures

The following 9 strains of NIVA-CHL1 were derived from 8 microalgal culture collections and one laboratory: ATCC 22662 from the American Type Culture Collection (ATCC), CCAP 278/4 from the Culture Collection of Algae and Protozoa (CCAP), CS-327-ANACC and CS-327-CSIRO from the Commonwealth Scientific and Industrial Research Organisation (CSIRO), NIES-35 from the Microbial Culture Collection at the National Institute for Environmental Studies (NIES), NIVA-CHL1 from the Norwegian Culture Collection of Algae (NORCCA), SAG 61.81 from the Culture Collection of Algae at Göttingen University (SAG), UTEX 1648 from the Culture Collection of Algae at The University of Texas at Austin (UTEX), and one strain from a laboratory in Japan (ATCC 22662–2), which has been maintained for approximately 20 years after it was received from ATCC (Table 1). The strains were propagated under continuous light (60–80 $\mu mol \cdot m^{-2} \cdot s^{-1}$) at 23°C in OECD medium [1]. Part of all nine cultures was cryopreserved as follows: cryoprotectant (5% DMSO, final concentration) was added to the culture, and the cells were cooled at a rate of −1°C/min to −35°C in a programmable freezer (Planer Kryo 360–1.7) and then frozen rapidly to –196°C in liquid nitrogen.

**Table 1.** *Raphidocelis subcapitata* strains used in this study.

| Location | Culture collection | Strain code |
|---|---|---|
| Japan | Microbial Culture Collection at the National Institute for Environmental Studies (NIES) | NIES-35 |
| | https://mcc.nies.go.jp/strainList.do?strainId=26&strainNumberEn=NIES-35 | |
| Scotland | Culture Collection of Algae and Protozoa (CCAP) | CCAP 278/4 |
| | https://www.ccap.ac.uk/strain_info.php?Strain_No=278/4 | |
| USA | Culture Collection of Algae at The University of Texas at Austin | UTEX 1648 |
| | https://utex.org/products/utex-1648 | |
| Australia | Commonwealth Scientific and Industrial Research Organisation (ANACC) | CS-327-ANACC |
| | https://www.csiro.au/en/Research/Collections/ANACC/Australian-National-Algae-Supply-service | |
| USA | American Type Culture Collection (ATCC) | ATCC 22662 |
| | https://www.atcc.org/products/all/22662.aspx?geo_country=us | |
| Norway | Norwegian Culture Collection of Algae (NORCCA) | NIVA-CHL1 |
| | https://niva-cca.no/shop/chlorophyceae/raphidocelis/niva-chl-1 | |
| Australia | Commonwealth Scientific and Industrial Research Organisation (CSIRO) | CS-327-CSIRO |
| | https://www.csiro.au/en/Research/Collections/ANACC/Australian-National-Algae-Supply-service | |
| Germany | Culture Collection of Algae at Göttingen University (SAG) | SAG 61.81 |
| | https://sagdb.uni-goettingen.de/detailedList.php?str_number=61.81 | |
| Japan | One laboratory in Japan | ATCC 22662–2 |
| | This strain was received from ATCC. | |

## Genome sequencing

DNA was extracted from 10 mL cultures with an Agencourt Chloropure kit (Beckman Coulter, Brea, USA) following the manufacturer's protocol and fragmented to approximately 550 bp using an M220 ultrasonicator (Covaris, Woburn, USA). Genomic libraries of paired-end reads were constructed and barcoded using an NEBNext Ultra II DNA Library Prep Kit for Illumina (New England Biolabs, Ipswich, USA) and NEBNext Multiplex Oligos for Illumina (Index Primers Set 1) (New England Biolabs). The library concentrations and size distributions were determined with NEBNext Library Quant Kit for Illumina (New England Biolabs) and Agilent 2200 Tape Station (Agilent Technologies, USA). Equinanomolar concentrations from each library were pooled. Next-generation sequencing was performed on a HiSeq X Ten system instrument as 150 bp paired-end reads by Novogene Corporation (Beijing, China) or on a MiSeq system instrument as 300 bp paired-end reads. Bases with a quality value of <20 were removed and reads shorter than 30 bases were discarded by using Sickle [6]. The filtered reads were mapped to the reference sequence of NIES-35 [3] in BWA ver.0.7.12 [7]. Duplicated reads were removed in Picard ver.2.0.1 (http://broadinstitute.github.io/picard/). The total number of reads obtained and the number of mapped reads for each strain are shown in Table 2. The data were deposited in DDBJ under accession number DRA010570 (DRR238908-DRR238916). Single-nucleotide polymorphisms (SNPs) and indels were detected in GATK ver.3.6 (https://github.com/broadinstitute/gatk/). The variant calling was performed according to the GATK best practices (http://software.broadinstitute.org/gatk/best-practices). Since there is no known variant data, base quality score recalibration was not performed. SNPs and indels were classified in snpEff ver.4.2 [8] (http://snpeff.sourceforge.net/SnpEff_manual.html).

## Phylogenetic analysis

Information on each SNP was extracted by ref_map.pl of Stacks ver.2.2 [9] with default settings, and then used to construct molecular phylogenetic trees by the maximum likelihood method in RAxML ver.8. 2. 9 with the GTR+$\Gamma$ model [10]. Branch support was evaluated with 100 bootstrap replicates. The tree file was visualized with pgsumtree in Phylogears2 ver.2.0.2015.11.30 (https://www.fifthdimension.jp/products/phylogears/).

## Growth inhibition tests

Two chemicals were used: 3,5-DCP (Lot: SDK3769, > 98.0%) and sodium chloride (NaCl) (Lot: 7935, >99.5%), both from Fujifilm Wako Pure Chemical Corporation (Osaka, Japan). Growth inhibition tests with these chemicals were performed according to OECD test guideline TG201 [1]. A preculture of axenic algae in OECD medium was started at least 6 days

**Table 2. Summary of genome sequencing using Illumina MiSeq.**

| Strain code | Number of reads | Total read bases | Number of mapped reads | % of mapped reads | Mean depth |
|---|---|---|---|---|---|
| NIES-35 | 7,942,335 | 3,613,278,372 | 7,893,009 | 99.4 | 38.1 |
| CCAP 278/4 | 38,103,886 | 11,245,438,309 | 34,603,513 | 90.8 | 68.0 |
| UTEX 1648 | 58,322,119 | 17,225,893,008 | 55,801,118 | 95.7 | 95.8 |
| CS-327-ANACC | 44,493,723 | 14,735,831,693 | 12,881,762 | 29.0 | 47.6 |
| ATCC 22662 | 45,928,502 | 13,546,010,659 | 43,072,168 | 93.9 | 79.6 |
| NIVA-CHL1 | 47,085452 | 14,961,236,143 | 9,832,384 | 20.8 | 33.8 |
| CS-327-CSIRO | 72,814,874 | 10,776,103,449 | 67,699,124 | 93.0 | 107.7 |
| ATCC 22662–2 | 49,731,566 | 14,685,552,895 | 46,883,978 | 94.4 | 85.4 |
| SAG 61.81 | 45,520,635 | 13,442,821,111 | 43,133,069 | 94.7 | 79.5 |

before the beginning of the test. Algae were harvested from batch culture during exponential growth. Algal suspensions (100 mL; initial concentration $5 \times 10^3$ cells mL$^{-1}$) containing the test chemicals (0.25, 0.5, 1.0, 2.0 and 4.0 mg/L in the testing of 3,5-DCP and 111, 333, 1000, 3000 and 9000 mg/L in the testing of NaCl as nominal concentration) were cultured in 300 mL Erlenmeyer flasks. Three replicates for each concentration and six replicates for the control sample were performed. The flasks were shaken continuously with orbital agitation of 100 rpm at 23 ± 1˚C under white fluorescent light (60–80 μmol·m$^{-2}$·s$^{-1}$). Test culture (1 mL) was collected every 24 h for up to 72 h after the start of exposure to toxicants, diluted to 10 mL with Cellpack (Sysmex, Kobe, Japan), and density and diameter of 3–12 μm cells were measured with a Sysmex CDA-500 electronic particle counter (Sysmex). The cell densities at 0, 24, 48, and 72 h were used to calculate the growth rate and effective concentrations. Toxicity values (NOEC, EC10, and EC50) were calculated with geometric average of measured concentrations at 0h and 72h of 3,5-DCP and nominal concentrations of NaCl. To measure the amount of 3,5-DCP in media, media at 0h and 72 after exposure were sampled (samples at 72h after exposure were filtered with 0.75 μm PTFE membrane filter to remove algae), and solid phase extraction was performed with OASIS HLB Plus cartridge (Waters, Milford, USA). Samples were extracted with methanol, and were analyzed with liquid chromatography-mass spectrometry (LCMS-2010EV; Shimazu, Kyoto, Japan).

## Statistical analysis

Differences in NOEC and EC values were analyzed in the open-source statistical software R (R: A language and environment for statistical computing. R Foundation for Statistical Computing, Vienna, Austria, http://www.R-project.org/). A one-tailed Dunnett's procedure was applied to estimate the NOEC and LOEC. The EC values were calculated by the 2-parameter log-logistic models in R with the extension package drc [11, 12]. The model function $f(x)$ shown in Eq I provides the relative endpoint response to concentration $x$, where the upper limit is fixed at the mean response in the control (100%) and $b$ and $e$ are parameters:

$$f(x; b, e, ) = 1/1 + \exp\left(b[\log(x) - \log(e)]\right), \tag{I}$$

## Results

Comparison of the nuclear genomes of NIVA-CHL1 maintained in different collections is shown in Table 2. The approximate number of high-quality paired reads ranged from approximately 8 million (NIES-35) to 73 million (CS-327-CSIRO), and the number of reads that were mapped to the reference genome of NIES-35 [3] ranged from approximately 8 million to 68 million. The mean coverage depth was 33.8–107.7; those of CS-327-ANACC and NIVA-CHL1 were relatively low because the two strains (Table 2) were not axenic. However, we considered

**Table 3. Summary of categorization of single-nucleotide polymorphisms (SNPs) and indels in comparison with the genome of NIES-35.**

| Strain code | Total number of SNPs and indels | HIGH | LOW | MODERATE | MODIFIER |
|---|---|---|---|---|---|
| CCAP 278/4 | 23,635 | 348 | 2,434 | 1,898 | 11,7191 |
| UTEX 1648 | 28,212 | 416 | 2,673 | 1,926 | 131,857 |
| CS-327-ANACC | 19,576 | 236 | 3,027 | 2,604 | 100,780 |
| ATCC 22662 | 25,666 | 362 | 2,553 | 1,921 | 123,684 |
| NIVA-CHL1 | 19,958 | 283 | 3,239 | 2,666 | 101,715 |
| CS-327-CSIRO | 27,939 | 392 | 2,611 | 1,953 | 131,083 |
| ATCC 22662–2 | 27,481 | 398 | 2,673 | 1,984 | 129,401 |
| SAG 61.81 | 26,242 | 344 | 2,521 | 1,901 | 125,519 |

the mapping efficiency of the obtained sequence data to be adequate for comparative genome analysis.

Each genome had SNPs or indels at 19,576 to 28,212 sites in comparison with the genome of NIES-35 (Table 3). Categorization of each SNP and indel using snpEff revealed that most of them were in non-coding regions such as introns (snpEff term: intron_variant) and 5′ and 3′ regions (snpEff terms: upstream_gene and downstream_gene variants); however, variants in coding regions of the genes, such as frameshift mutations (snpEff term: frameshift variant) and nonsense mutations (snpEff term: stop_gained variant), were also found and had various consequences (Fig 1, Table 3).

We categorized the variants into 4 classes according to potential phenotypes: impact HIGH, variants that drastically change protein function such as "stop gained" and "frameshift"; impact LOW, variants that possibly change protein function such as synonymous SNPs; impact MODERATE, variants that hardly affect protein function such as missense mutations (snpEff term: missense variant) and in-frame mutations (snpEff terms: inframe_deletion and insertion variant); and impact MODIFIER, variants in intergenic regions that do not affect protein function. Variants at 236–416 sites were classified as impact HIGH, variants at 2434–3239 sites as impact LOW, variants at 1898–2666 sites as impact MODERATE, and variants at 100,780–131,857 sites as impact MODIFIER (Fig 2 and Table 3). Classification of proteins with "impact HIGH" variants using Clusters of Orthologous Groups of Proteins (COGs) showed no enrichment of variants from specific gene families (Fig 2B). However, the proportion of each category was somewhat different among the strains. For example, the proportion of category A was lower in CS-327-ANACC (2.5%) than in the other strains (4.4%–6.4%), and that of category C was lower in ATCC 22662 and CCAP 278/4 (2.7%) than in the other strains (4.4%–7.1%) (Fig 2B).

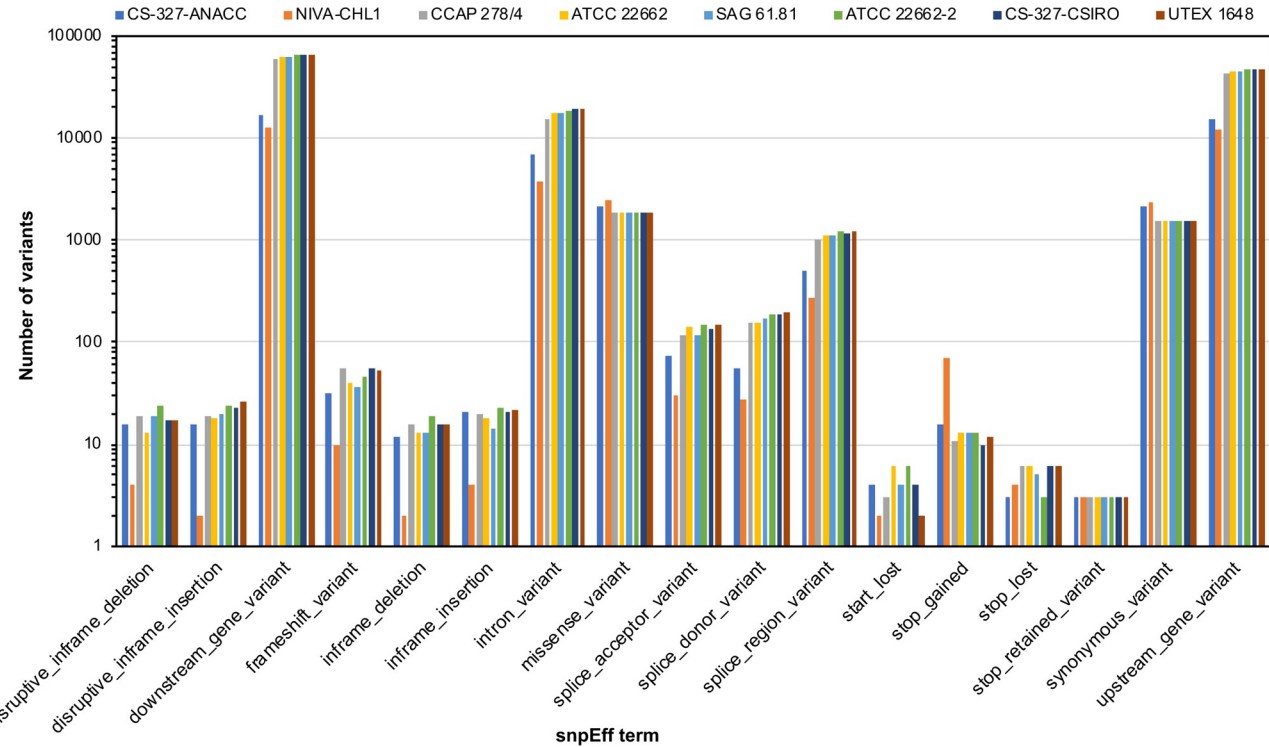

**Fig 1. Categorization of each single-nucleotide polymorphism and indel by snpEff term.** The terms are explained at http://snpeff.sourceforge.net/snpEff_manual.html.

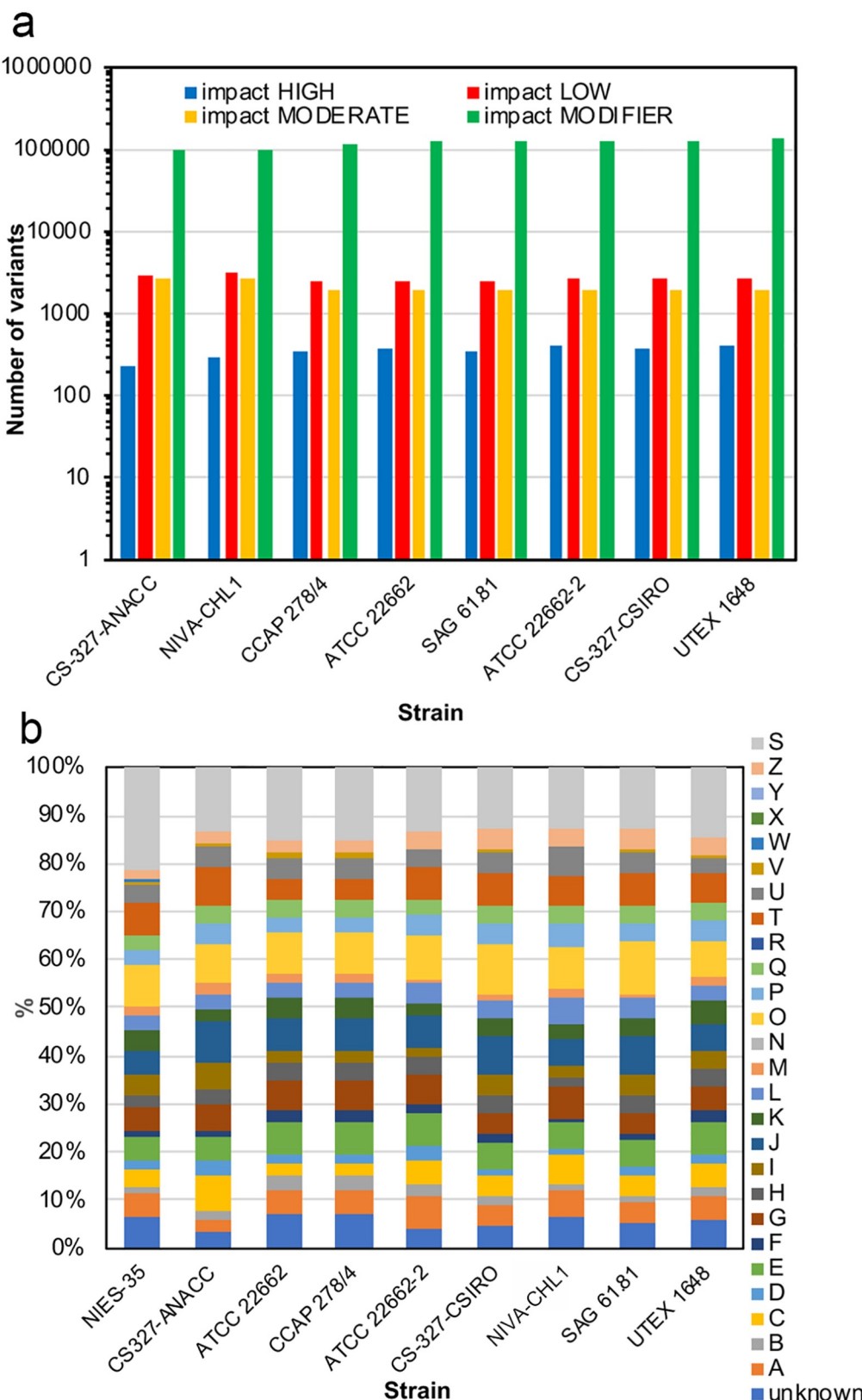

**Fig 2. Classification of variants by their potential to produce a phenotype, and functional classification of proteins with variants.** (a) Variants were classified into 4 impact categories (HIGH, LOW, MODERATE, and MODIFIER) using snpEff. (b) Proteins with "impact HIGH" variants were functionally classified using Clusters of Orthologous Groups of Proteins (COGs). Uppercase letters are explained at https://www.ncbi.nlm.nih.gov/COG/.

Phylogenetic analysis based on SNP information showed that NIVA-CHL1 was in the same clade with NIES-35 and CS-327-ANACC, which was supported by a high bootstrap value (100%) (Fig 3). As expected, ATCC 22662 was grouped with ATCC 22662–2, a strain from a laboratory in Japan, because the latter was delivered from ATCC approximately 20 years ago. However, the branch of each of the two strains was very long (Fig 3), indicating that their genomes have diverged considerably over 20 years.

For growth inhibition tests to assess the variation in sensitivity among the 9 strains, we used 3,5-DCP because it is a typical reference substance. All the 9 strains fulfilled the criteria of OECD TG201 that the biomass in the control culture should increase exponentially by a factor of at least 16 within a 72-h test period. However, each factor was different among the 9 strains: the biomass of NIES-35 increased 544-fold in 72 h, whereas that of CS-327-CSIRO increased by only 66-fold. For CS-327-ANACC, the EC10 (0.38 mg/L) and EC50 (0.85 mg/L) values were the lowest among the 9 strains, whereas for SAG 61.81, the EC10 (1.17 mg/L) and EC50 (2.22 mg/L) values were the highest (Fig 4, Table 4).

The gene encoding small conductance mechanosensitive (MS) ion channel (Rsub_08752) (Fig 5A) had a one-base substitution at position 2667 (adenine to cytosine) in ATCC 22662–2 and CCAP 278/4 (Fig 5B and 5C). This position was predicted to be a splice acceptor site (two bases before exon start). Therefore, in ATCC 22662–2 and CCAP 278/4, this mutation causes the MS ion channel to be truncated, resulting in loss of function or nonsense-mediated decay; thus, the ability to adapt to external osmotic stress might be changed. Also, a prediction of the presence of signal peptides by ChloroP (http://www.cbs.dtu.dk/services/ChloroP/) evaluated

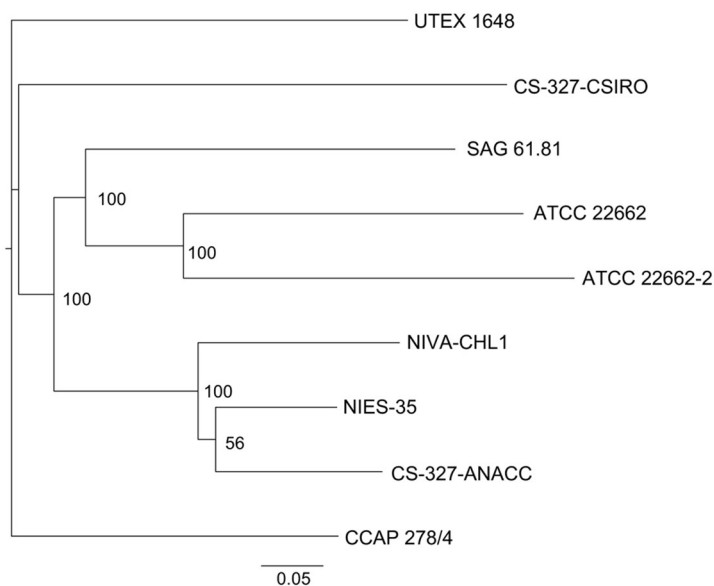

**Fig 3. Unrooted molecular phylogenetic tree based on single-nucleotide polymorphisms.** The tree was constructed using the maximum likelihood method with RAxML (GTR+Γ). Bootstrap values (if higher than 50%) are given along the branches. Scale bar shows the number of substitutions per site.

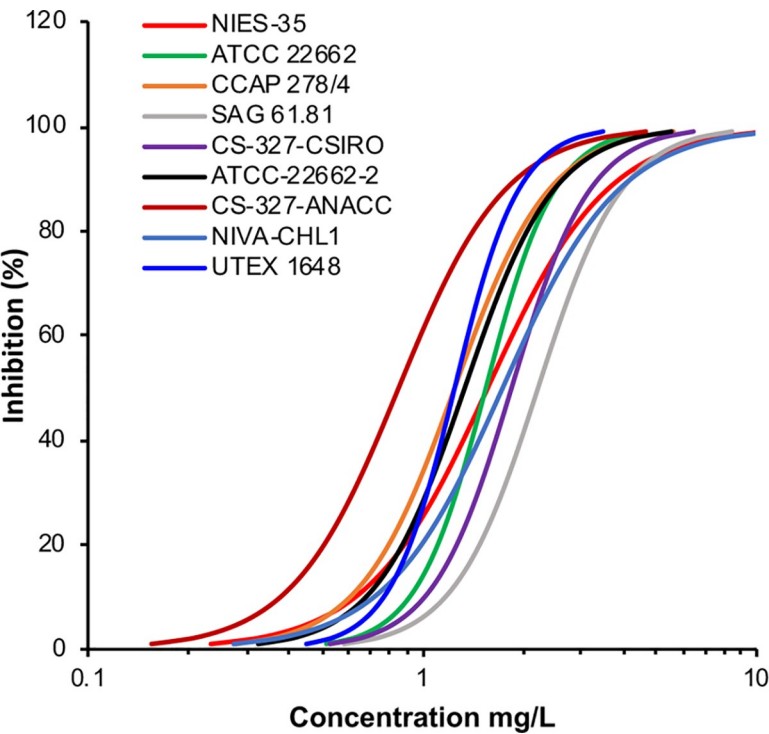

**Fig 4. Dose-response curves based on the data from the growth inhibition test with 3,5-dichlorophenol.**

that Rsub_08752 protein possesses a chloroplast transit peptide of 20 amino acids at the N-terminus, indicating that it might localize at the chloroplast envelope (Fig 5A).

To evaluate whether this mutation affects the ability to adapt to external osmotic stress, we used a growth inhibition test with NaCl. The growth rate of ATCC 22662–2 and CCAP 278/4 tended to be lower than that of NIES-35 in media without NaCl and with 111 mg/L NaCl; the difference became significant at 333 and 1000 mg/L NaCl (Fig 6A and 6B). At 3000 mg/L, the growth of all three strains was strongly inhibited, with that of NIES-35 being the lowest (Fig 6A), indicating that ATCC 22662–2 and CCAP 278/4 became more tolerant to external osmotic stress than NIES-35. The growth rate of ATCC 22662–2 and CCAP 278/4 increased slightly but significantly in medium containing 333 mg/L NaCl (Fig 6A). The EC50 values of NaCl for

**Table 4. Sensitivity to 3,5-dichlorophenol.**

|  | NOEC mg/L | EC$_{10}$ mg/L | 95% confidence limit | EC$_{50}$ mg/L | 95% confidence limit |
|---|---|---|---|---|---|
| **NIES-35** | < 0.25 | 0.63 | 0.47–0.79 | 1.57 | 1.42–1.72 |
| **CCAP 278/4** | 0.23 | 0.61 | 0.54–19.4 | 1.25 | 1.19–1.31 |
| **UTEX 1648** | 0.39 | 0.77 | 0.72–0.82 | 1.25 | 1.20–1.30 |
| **CS-327-ANACC** | <0.18 | 0.38 | 0.35–0.41 | 0.85 | 0.82–0.88 |
| **ATCC 22662** | 0.44 | 0.92 | 0.84–0.10 | 1.55 | 1.50–1.61 |
| **NIVA-CHL1** | <0.20 | 0.72 | 0.40–1.03 | 1.74 | 1.50–1.99 |
| **CS-327-CSIRO** | <0.39 | 0.81 | 0.74–0.89 | 1.49 | 1.43–1.54 |
| **ATCC 22662–2** | 0.23 | 0.68 | 0.59–0.77 | 1.34 | 1.26–1.41 |
| **SAG 61.81** | 0.19 | 1.17 | 1.07–1.27 | 2.22 | 2.12–2.31 |

Values were calculated with measured concentrations.

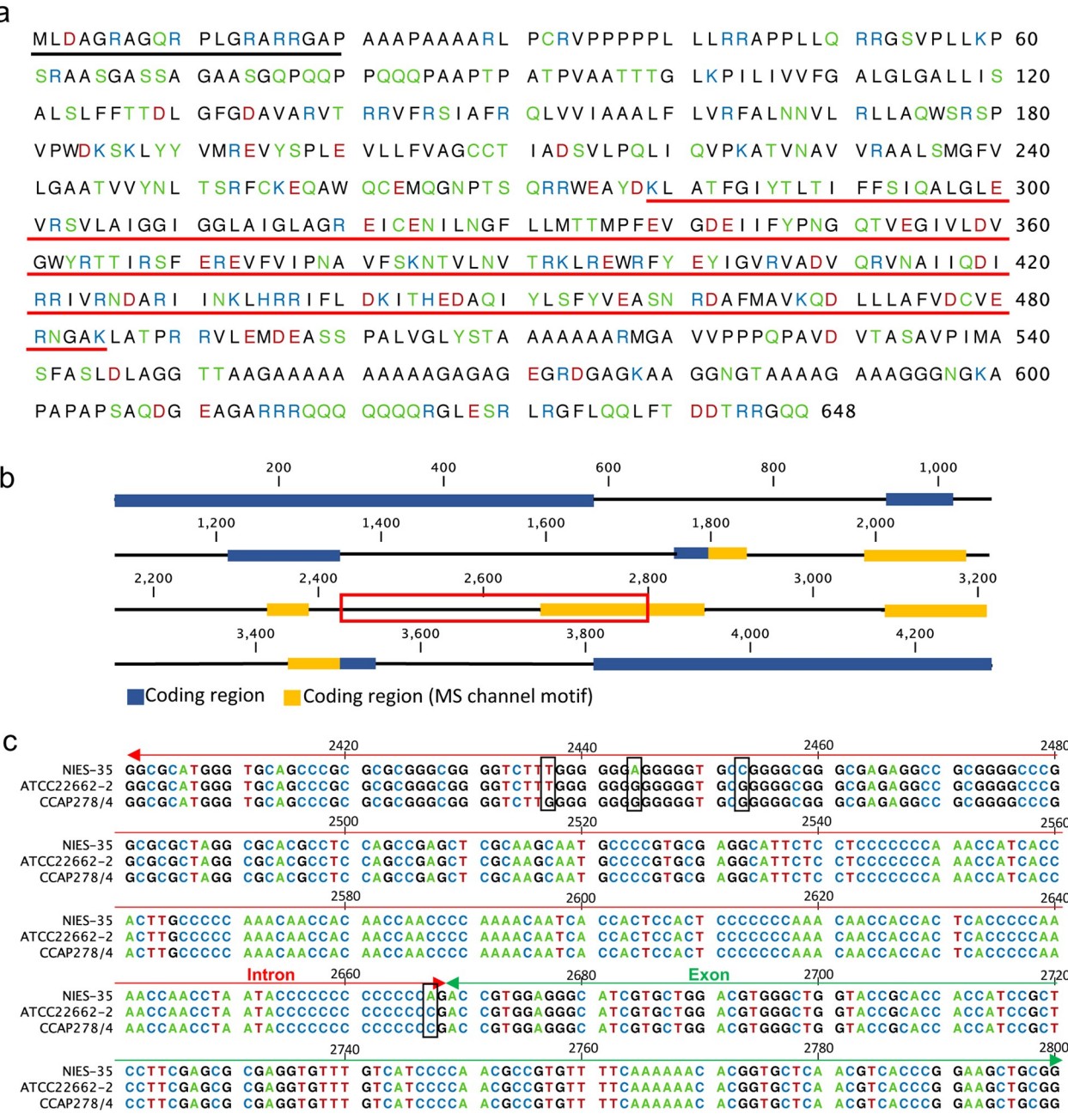

**Fig 5. Predicted amino acid sequence of small-conductance mechanosensitive (MS) ion channel in strains derived from NIVA-CHL1 and schematic representation of its gene.** (a) The predicted amino acid sequence; amino acid numbers beginning from the initiator methionine are shown on the right. Black underline, signal peptide; red underline, MS channel motif. (b) Structure of the gene (Rsub_08752) encoding MS ion channel. Boxes, exons; lines, introns. (c) Alignment of DNA sequences enclosed in the red box in (b). Single-nucleotide polymorphisms are shown by black boxes.

ATCC 22662–2 (2304 mg/L) and CCAP 278/4 (2497 mg/L) were higher than that of NIES-35 (1600 mg/L) (Fig 6C). To evaluate the ability to adapt to external osmotic stress, we compared cell diameter at 72 h after the onset of NaCl exposure among the three strains (Fig 6D). Cell diameter tended to be larger in ATCC 22662–2 and CCAP 278/4 than in NIES-35 in the control

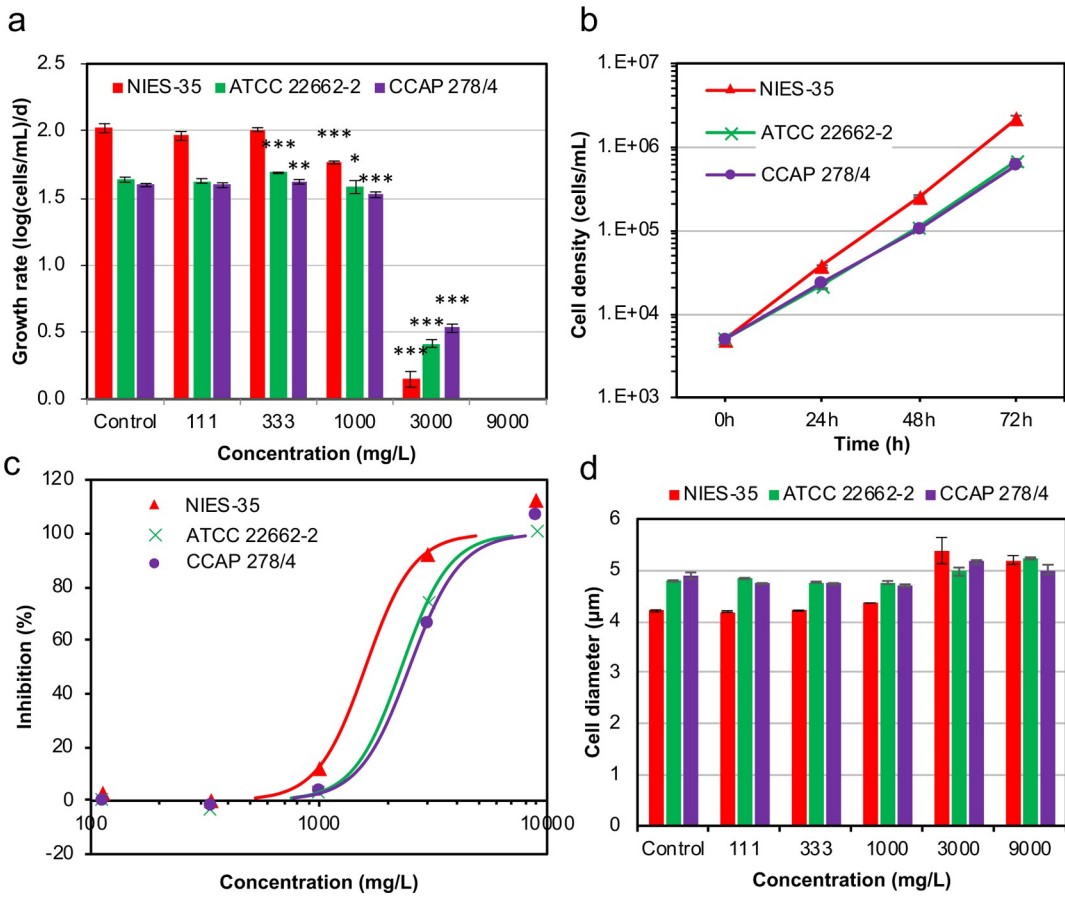

**Fig 6. Growth inhibition test of NIES-35, ATCC 22662–2, and CCAP 278/4 with NaCl.** (a) Specific growth rate (averaged over 0–24, 24–48, and 48–72 h). *** $p < 0.001$; ** $p < 0.01$; * $p < 0.1$ vs. control without NaCl. (b) Cell densities in medium without NaCl (Control). (c) Dose-response curves based on the growth inhibition data. (d) Average cell diameter at 72 h after the onset of exposure. Error bars in (a), (b), and (d) denote standard deviation.

culture. Cell diameter of ATCC 22662–2 and CCAP 278/4 was larger than that of NIES-35 but decreased slightly in media containing 111, 333, and 1000 mg/L NaCl (Fig 6D). A remarkable increase in cell diameter was found in media containing 3000 or 9000 mg/L in all three strains; at these NaCl concentrations, no clear differences among the strains were observed (Fig 6D).

## Discussion

We found that the genome of the test algal strain *R. subcapitata* NIVA-CHL1 varied greatly, with many SNPs and indels, among algal culture collections, even though the strains in these collections were originally derived from the same NIVA-CHL1 culture. These mutations were detected not only in non-coding but also in coding regions, where they may affect protein function. Mutations in coding regions are a critical issue for test algal species because they may affect the sensitivity to chemical substances. Actually, the EC50 values in the growth inhibition test with 3,5-DCP varied up to 2.6-fold among the 9 strains. This variability was too large for a test performed in the same laboratory: if a test was performed with different strains in the different laboratories, the difference in sensitivity would be even larger because a result is greatly affected by the differences of handing and test procedure. Thus, we suggest that this difference in sensitivity resulted from genomic mutations. Although it would be very interesting to

establish which gene is involved in this difference, this is rather hard to predict because the mode of action of 3,5-DCP in algae is unclear.

We found that the MS ion channel gene has an SNP that may cause protein truncation and loss of function or trigger nonsense-mediated decay in ATCC 22662–2 and CCAP 278/4. In *Escherichia coli*, the MS ion channel is intrinsically opened in response to stretching of the cell membrane without the participation of cytoskeletal or other components [13, 14]; therefore, it directly mediates the adaptation to osmotic stress and regulates cell volume. Large-conductance MS ion channel (MscL) and small-conductance MS ion channel (MscS) have been identified in the cytoplasmic membrane of *E. coli*, and the function of these channels has been characterized using the patch clamp technique [13–20]. Despite similar functions, the genes encoding the two MS channels are not homologous, indicating a different origin [13]. MscL is present throughout prokaryotes, whereas MscS homologs are also found throughout eukaryotes [13]. In the land plant *Arabidopsis thaliana*, MS Channel of Small Conductance-Like2 and 3 (MSL2 and MSL3) localize in the plastid envelope [21]. Using an *msl2 msl3 Arabidopsis* mutant, these authors demonstrated that MSL2 and MSL3 are involved in plastid osmoregulation, and plastid osmotic stress activates cellular stress responses. They also showed that the *msl2 msl3* mutant contains large and swollen plastids in comparison with those in the wild type and has defects in leaf morphology and growth because of a defect in cellular osmoregulation, but these phenotypes recover in NaCl-containing media. We did not examine plastid morphology of ATCC 22662–2 and CCAP 278/4, but found that their slower growth in medium without NaCl in comparison with NIES-35 recovered significantly at 333 mg/L NaCl. Also, an increase in cell diameter slightly recovered at 111, 333, or 1000 mg/L NaCl in CCAP 278/4. We also considered that increased tolerance to external osmotic stress in ATCC 22662–2 and CCAP 278/4 results from the recovery of growth rate in NaCl-containing media, as in the *msl2 msl3* mutant. These results suggest that the ability to adapt to external osmotic stress was changed by the loss of function of MscS in ATCC 22662–2 and CCAP 278/4. Because the *R. subcapitata* genome encodes Rsub_5425 (a homolog of Rsub_08752), in which no critical mutation was detected, a double mutation of both genes might result in a stronger phenotype than that of a single mutant. We also detected critical mutations in 9–15 genes involved in inorganic ion transport such as metal ABC transporter permease and metal-nicotianamine transporter-like in all of the 9 strains. Tolerance of these strains to heavy metals and their requirements for trace metals should be analyzed in the future. Thus, the study suggests that genomic alterations in NIVA-CHL1 may result in considerable variation in toxicity values among the strains maintained in different culture collections, especially in the case of a mutation in the target of a test chemical substance.

The main reason that different variants developed in the culture collections is probably long-term sub-culturing under different conditions. For example, at NIES, the strain was maintained under a 10-h light/14-h dark photoperiod (4–10 μmol photons m$^{-2}$s$^{-1}$) at 20˚C in C medium [22] until 2005. On the other hand, at UTEX, the strain was maintained under a 12-h light/12-h dark photoperiod (approximately 13 μmol photons m$^{-2}$s$^{-1}$) at 20˚C in Bristol medium [23] from 1968 to 2006. Some mutations might contribute to the gain of useful traits or loss of useless traits under certain culture conditions, as demonstrated in *Drosophila*: genes encoding an olfactory receptor and a light receptor were mutated by maintaining the flies in the dark for 57 years. In *Chlamydomonas reinhardtii* that is closely related in *R. subcapitata*, the strain kept in different laboratory lost nitrate reductase very early in the strain history [24], and METE (Cobalamin-independent methionine synthase) mutant was able to be generated in conditions of high vitamin B12 concentration over ca.500 cell generations [25]. It is easy to predict that environmental adaptive traits such as MscS activity involved in osmoregulation might be useless under constant artificial culture conditions.

It is urgent to standardize the maintenance method of the algal strain among culture collections to prevent or slow down divergent mutagenesis. Cryopreservation would be the most desirable method for long-term maintenance because it is not time consuming and prevents contamination and genetic drift. To the best of our knowledge, only ATCC, NIES, SAG, UTEX and CCAP, and to a lesser degree ANACC already use cryopreservation to maintain algal strains. It is also necessary to re-select any one of the NIVA-CHL1 strains as the standard one. We propose to select the new standard strain based on further researches about the perspective of the degree of mutagenesis: it is important to know each strain exhibits what kind of phenotypes related in the genome mutations, and whether there are any critical mutations affecting results of the growth inhibition tests.

## Conclusions

In this study, we found that the genome of the test algal strain *R. subcapitata* NIVA-CHL1 varied greatly, with many SNPs and indels, among the 9 strains maintained at algal culture collections worldwide or testing laboratory. The sensitivity to 3,5-DCP varied among the 9 strains, and the ability to adapt to external osmotic stress was changed probably by the loss of function of MscS in ATCC 22662–2 and CCAP 278/4. These results suggest that genomic alterations in NIVA-CHL1 may result in considerable variation in sensitivity values against potential toxins among the strains maintained in different culture collections.

## Acknowledgments

We thank M. Koshio for the quantitative analyses of 3,5-DCP and A. Kariya and M. Tayama for their great effort and support in algal growth inhibition testing.

## Author Contributions

**Conceptualization:** Takahiro Yamagishi.

**Formal analysis:** Takahiro Yamagishi, Haruyo Yamaguchi, Shigekatsu Suzuki, Mayumi Yoshikawa.

**Resources:** Ian Jameson, Maike Lorenz, David R. Nobles, Christine Campbell.

**Supervision:** Masanori Seki, Masanobu Kawachi, Hiroshi Yamamoto.

**Writing – original draft:** Takahiro Yamagishi.

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
