## [Decision Letter · Decision Letter 0]

25 Sep 2020

PONE-D-20-26198

Comparative genome analysis of test algal strain NIVA-CHL1 (*Raphidocelis subcapitata*) maintained in microalgal culture collections worldwide

PLOS ONE

Dear Dr. Yamagishi,

Thank you for submitting your manuscript to PLOS ONE. After careful consideration, we feel that it has merit but does not fully meet PLOS ONE’s publication criteria as it currently stands. Therefore, we invite you to submit a revised version of the manuscript that addresses the points raised during the review process.

We look forward to receiving your revised manuscript.

Kind regards,

Yi Cao

Academic Editor

PLOS ONE

Journal Requirements:

'This research is partially supported by the National BioResource Project (NBRP) from Japan Agency for Medical Research and Development (AMED).

'The author(s) received no specific funding for this work.'

Please ensure your amended statement declares *all* the funding or sources of support (whether external or internal to your organization) received during this study, as detailed online in our guide for authors at http://journals.plos.org/plosone/s/submit-now

If partially funded, please also include the statement “There was no additional external funding received for this study.” in your updated Funding Statement.

Reviewers' comments:

Reviewer's Responses to Questions

**Comments to the Author**

1. Is the manuscript technically sound, and do the data support the conclusions?

Reviewer #1: Yes

Reviewer #2: Yes

2. Has the statistical analysis been performed appropriately and rigorously? 

Reviewer #1: Yes

Reviewer #2: Yes

3. Have the authors made all data underlying the findings in their manuscript fully available?

Reviewer #1: Yes

Reviewer #2: Yes

4. Is the manuscript presented in an intelligible fashion and written in standard English?

Reviewer #1: Yes

Reviewer #2: No

5. Review Comments to the Author

Reviewer #1: Thank you for the opportunity to read and review this manuscript. The authors have performed a comparative study looking into the genetic variation of 9 culture collection strains of Raphidocelis subcapitata, all of which were derived from the same isolated strain NIVA-CHL1. Variants (SNPs and indels) were called in each strain relative to the reference genome and were assessed for their potential impacts on protein function. Two strains (ATCC 22662-2 and CCAP 278/4) contained mutations in an ion channel protein and growth inhibition tests with NaCl indicated that these mutations might influence osmotic regulation. All 9 strains additionally showed vast differences in toxicant sensitivity. The authors suggest that these differences may be due to acquired mutations from long-term sub-culturing under different conditions.

The authors present a very interesting study that has clear implications for those studying microalgal strains housed in culture collections. It should not be surprising that genetic variation gets acquired during long-term culturing under different conditions. The results presented here offer convincing evidence that laboratory cultures of the same strain act differently. I would also argue that these points about random genetic mutation should be taken into consideration for all labs that work with model organisms. The authors point out the important need to standardize the maintenance of model algal strains and that cryopreservation offers some benefit.

Overall, I found this manuscript to be well-written and, given the stated goals of the study, the results are convincing and presented well. The methods were appropriate. The figures were useful and designed well. Many of my suggestions that I detail below are to clarify or provide additional details about how some of the methods were performed.

Suggestions:

Line 140: Provide information about whether the libraries were barcoded (single or dual indices) and any pooling details.

Line 143: What was the length of the sequenced reads produced from the HiSeq X Ten and MiSeq systems? Were they single- or paired-end reads?

Line 150: More details should be included about variant calling in GATK. For instance, was the GATK Best Practices followed (https://www.broadinstitute.org/partnerships/education/broade/best-practices-variant-calling-gatk-1), or a modified pipeline? Was indel realignment performed? Was base quality score recalibration performed? Were the initial variants filtered for quality or depth?

Line 254: I would add here in parentheses what the bootstrap values were for these relationships that you are describing.

Line 328: “sensitivity of chemical substances” to “sensitivity to chemical substances.”

Line 364: “in genes of 9-15 genes” to “in 9-15 genes”

Reviewer #2: Generally, Raphidocelis subcapitata was one of the most frequently used species for algal growth inhibition tests. Therefore, genetic characteristics of R. subcapitata was very crucial for its functional applications. This manuscript compared the genome sequences among NIVA-CHL1 from 9 microalgal culture collections to evaluate the presence of mutations. The results was very meaningful for subsequent applications by different mutations. However, some issues need to be addressed as the following point to point:

1) In Introduction Section, the author should describe the significance of this study in investigating mutants.

2) In Materials and Methods Section (line 131), the strains were propagated under continuous light at 23°C in OECD medium. The illumination intensity and the compositions in OECD medium should be shown.

3) In line 138, 10-mL should be changed into 10 mL, please unify this similar situation in the whole manuscript.

4) In line 186, it would be better to note the source of the species by references in Table 1.

5) In the Growth inhibition tests, two chemicals were used: 3,5-DCP (Lot: SDK3769, > 98.0%) and sodium chloride (NaCl) (Lot: 7935, >99.5%), however, the concentrations of the two chemicals in the medium were not mentioned. Please described in detail.

6) Please carefully check grammatical and typing errors in the whole manuscript and please provide a higher resolution data graphs.

In sum, I recommend publication of the paper in PLOS ONE, provided the authors comply with my comments above.

6. PLOS authors have the option to publish the peer review history of their article (what does this mean?). If published, this will include your full peer review and any attached files.

Reviewer #1: No

Reviewer #2: No

---

## [Author Response · Author response to Decision Letter 0]

15 Oct 2020

Dear editors,

Thank you very much for your efforts in editing our submission. We revised the manuscript considering the valuable comments by you and the reviewers. Regarding one major point of the suggestions from Reviewer 1 on providing additional information how the genome sequence and analyses were performed, we revised the manuscript adding more information about this. Also, we sincerely apologize for the grammatical and careless typographical errors in original submission. We have made a thorough check of this revised version, and hope that no errors remain. 

In addition, we responded to all of Journal Requirements on style, funding information in Acknowledgements, and so on. Please see the following, indicating our responses.

Review Comments to the Author

Reviewer #1: Thank you for the opportunity to read and review this manuscript. The authors have performed a comparative study looking into the genetic variation of 9 culture collection strains of Raphidocelis subcapitata, all of which were derived from the same isolated strain NIVA-CHL1. Variants (SNPs and indels) were called in each strain relative to the reference genome and were assessed for their potential impacts on protein function. Two strains (ATCC 22662-2 and CCAP 278/4) contained mutations in an ion channel protein and growth inhibition tests with NaCl indicated that these mutations might influence osmotic regulation. All 9 strains additionally showed vast differences in toxicant sensitivity. The authors suggest that these differences may be due to acquired mutations from long-term sub-culturing under different conditions.

The authors present a very interesting study that has clear implications for those studying microalgal strains housed in culture collections. It should not be surprising that genetic variation gets acquired during long-term culturing under different conditions. The results presented here offer convincing evidence that laboratory cultures of the same strain act differently. I would also argue that these points about random genetic mutation should be taken into consideration for all labs that work with model organisms. The authors point out the important need to standardize the maintenance of model algal strains and that cryopreservation offers some benefit.

Overall, I found this manuscript to be well-written and, given the stated goals of the study, the results are convincing and presented well. The methods were appropriate. The figures were useful and designed well. Many of my suggestions that I detail below are to clarify or provide additional details about how some of the methods were performed.

Suggestions:

Line 140: Provide information about whether the libraries were barcoded (single or dual indices) and any pooling details.

 Considering this comment, we revised as follows: “Genomic libraries of paired-end reads were constructed and barcoded using an NEBNext Ultra II DNA Library Prep Kit for Illumina (New England Biolabs, Ipswich, USA) and……”(p.7, line 144-146).

Line 143: What was the length of the sequenced reads produced from the HiSeq X Ten and MiSeq systems? Were they single- or paired-end reads?

 Considering this comment, we revised as follows: “Genomic libraries of paired-end reads were constructed and barcoded using an NEBNext Ultra II DNA Library Prep Kit for Illumina (New England Biolabs, Ipswich, USA) and NEBNext Multiplex Oligos for Illumina (Index Primers Set 1) (New England Biolabs). The library concentrations and size distributions were determined with NEBNext Library Quant Kit for Illumina (New England Biolabs) and Agilent 2200 Tape Station (Agilent Technologies, USA). Equinanomolar concentrations from each library were pooled. Next-generation sequencing was performed on a HiSeq X Ten system instrument as 150 bp paired-end reads by Novogene Corporation (Beijing, China) or on a MiSeq system instrument as 300 bp paired-end reads.”(p. 7-8, line 144-153).

Line 150: More details should be included about variant calling in GATK. For instance, was the GATK Best Practices followed (https://www.broadinstitute.org/partnerships/education/broade/best-practices-variant-calling-gatk-1), or a modified pipeline? Was indel realignment performed? Was base quality score recalibration performed? Were the initial variants filtered for quality or depth?

 Considering this comment, we revised as follows: “The variant calling was performed according to the GATK best practices (http://software.broadinstitute.org/gatk/best-practices). Since there is no known variant data, base quality score recalibration was not performed. SNPs and indels were classified in snpEff ver.4.2 [7](http://snpeff.sourceforge.net/SnpEff_manual.html).”(p. 8, line 161-164).

Line 254: I would add here in parentheses what the bootstrap values were for these relationships that you are describing.

 We added the bootstrap value (p. 15, line 270).

Line 328: “sensitivity of chemical substances” to “sensitivity to chemical substances.”

 We corrected this (p. 19, line 343).

Line 364: “in genes of 9-15 genes” to “in 9-15 genes”

 We corrected this (p. 20, line 386).

Reviewer #2: Generally, Raphidocelis subcapitata was one of the most frequently used species for algal growth inhibition tests. Therefore, genetic characteristics of R. subcapitata was very crucial for its functional applications. This manuscript compared the genome sequences among NIVA-CHL1 from 9 microalgal culture collections to evaluate the presence of mutations. The results was very meaningful for subsequent applications by different mutations. However, some issues need to be addressed as the following point to point:

1) In Introduction Section, the author should describe the significance of this study in investigating mutants.

 Considering this comment, we added the following sentence: “In addition, it is critical issue for the standard strain whether sensitivities to toxicants vary among the strains maintained at culture collections worldwide.”(p. 6, line 110-111)

2) In Materials and Methods Section (line 131), the strains were propagated under continuous light at 23°C in OECD medium. The illumination intensity and the compositions in OECD medium should be shown.

 Considering this comment, we revised as follows: “The strains were propagated under continuous light (60–80 μmol·m−2·s−1) at 23°C in OECD medium [1].”(p. 7, line 134)

3) In line 138, 10-mL should be changed into 10 mL, please unify this similar situation in the whole manuscript.

 We corrected this (p. 7, line 142).

4) In line 186, it would be better to note the source of the species by references in Table 1.

 Considering this comment, we added URL of webpage for the strain information in Table 1 (p. 10). 

5) In the Growth inhibition tests, two chemicals were used: 3,5-DCP (Lot: SDK3769, > 98.0%) and sodium chloride (NaCl) (Lot: 7935, >99.5%), however, the concentrations of the two chemicals in the medium were not mentioned. Please described in detail.

 Considering this comment, we added the following sentence: “(0.25, 0.5, 1.0, 2.0 and 4.0 mg/L in the testing of 3,5-DCP and 111, 333, 1000, 3000 and 9000 mg/L in the testing of NaCl as nominal concentration)” (p. 9, line 182-183)

6) Please carefully check grammatical and typing errors in the whole manuscript and please provide a higher resolution data graphs.

 We have made a thorough check of this revised version, and provided high resolution figures (TIFF: 300 dpi) to the editorial office together with the revised manuscript.

In sum, I recommend publication of the paper in PLOS ONE, provided the authors comply with my comments above.

Journal Requirements:

 We have made a thorough check of this revised version. 

 Regarding the phrase including “data not shown”, we revised as follows: “In our previous study, our alignment of the rbcL regions between NIES-35 and ATCC 22662 showed a mismatch of two base pairs in the rbcL gene [5] (p. 5. Line 100-102)” and “This variability was too large for a test performed in the same laboratory: if a test was performed with different strains in the different laboratories, the difference in sensitivity would be even larger because a result is greatly affected by the differences of handing and test procedure. Thus, we suggest that this difference in sensitivity resulted from genomic mutations (Page 19, line 343-347)”.

'This research is partially supported by the National BioResource Project (NBRP) from Japan Agency for Medical Research and Development (AMED).

 I removed any funding-related text from the manuscript, and included our updated statement in our cover letter.

 We checked this.

We hope that these revisions are satisfactory.

Sincerely Yours,

Takahiro Yamagishi

---

## [Decision Letter · Decision Letter 1]

22 Oct 2020

Comparative genome analysis of test algal strain NIVA-CHL1 (*Raphidocelis subcapitata*) maintained in microalgal culture collections worldwide

PONE-D-20-26198R1

Dear Dr. Yamagishi,

We’re pleased to inform you that your manuscript has been judged scientifically suitable for publication and will be formally accepted for publication once it meets all outstanding technical requirements.

Kind regards,

Yi Cao

Academic Editor

PLOS ONE

Additional Editor Comments (optional):

Reviewers' comments:

Reviewer's Responses to Questions

**Comments to the Author**

1. If the authors have adequately addressed your comments raised in a previous round of review and you feel that this manuscript is now acceptable for publication, you may indicate that here to bypass the “Comments to the Author” section, enter your conflict of interest statement in the “Confidential to Editor” section, and submit your "Accept" recommendation.

Reviewer #1: All comments have been addressed

Reviewer #2: All comments have been addressed

2. Is the manuscript technically sound, and do the data support the conclusions?

Reviewer #1: Yes

Reviewer #2: Yes

3. Has the statistical analysis been performed appropriately and rigorously? 

Reviewer #1: Yes

Reviewer #2: Yes

4. Have the authors made all data underlying the findings in their manuscript fully available?

Reviewer #1: Yes

Reviewer #2: Yes

5. Is the manuscript presented in an intelligible fashion and written in standard English?

Reviewer #1: Yes

Reviewer #2: Yes

6. Review Comments to the Author

Reviewer #1: Thank you to the authors for considering all comments and making the requested changes to the manuscript. I have read the revised manuscript and have no further comments or suggestions.

Reviewer #2: The author has answered and addressed all of my concerns, so I recommend publication in this journal.

7. PLOS authors have the option to publish the peer review history of their article (what does this mean?). If published, this will include your full peer review and any attached files.

Reviewer #1: No

Reviewer #2: No

---

## [Editor Report · Acceptance letter]

28 Oct 2020

PONE-D-20-26198R1 

Comparative genome analysis of test algal strain NIVA-CHL1 (*Raphidocelis subcapitata*) maintained in microalgal culture collections worldwide 

Dear Dr. Yamagishi:

I'm pleased to inform you that your manuscript has been deemed suitable for publication in PLOS ONE. Congratulations! Your manuscript is now with our production department. 

Kind regards, 

on behalf of

Dr. Yi Cao 

Academic Editor

PLOS ONE